# Functional Impact of BeKm-1, a High-Affinity hERG Blocker, on Cardiomyocytes Derived from Human-Induced Pluripotent Stem Cells

**DOI:** 10.3390/ijms21197167

**Published:** 2020-09-28

**Authors:** Stephan De Waard, Jérôme Montnach, Barbara Ribeiro, Sébastien Nicolas, Virginie Forest, Flavien Charpentier, Matteo Elia Mangoni, Nathalie Gaborit, Michel Ronjat, Gildas Loussouarn, Patricia Lemarchand, Michel De Waard

**Affiliations:** 1L’institut du thorax, INSERM, CNRS, Université de Nantes, F-44007 Nantes, France; stephan.dewaard@univ-nantes.fr (S.D.W.); jerome.montnach@univ-nantes.fr (J.M.); barbara.ribeiro@univ-nantes.fr (B.R.); sebastien.nicolas@univ-nantes.fr (S.N.); virginie.forest@univ-nantes.fr (V.F.); flavien.charpentier@univ-nantes.fr (F.C.); nathalie.gaborit@univ-nantes.fr (N.G.); michel.ronjat@univ-nantes.fr (M.R.); gildas.loussouarn@univ-nantes.fr (G.L.); patricia.lemarchand@univ-nantes.fr (P.L.); 2LabEx Ion Channels, Science & Therapeutics, F-06560 Valbonne, France; matteo.mangoni@igf.cnrs.fr; 3Institut de Génomique Fonctionnelle, CNRS, INSERM, Université de Montpellier, F34094 Montpellier, France; 4Smartox Biotechnology, 6 rue des Platanes, F-38120 Saint-Egrève, France

**Keywords:** BeKm-1, hERG, LQTS, hiPS-cardiomyocytes

## Abstract

I_Kr_ current, a major component of cardiac repolarization, is mediated by human *Ether-à-go-go*-Related Gene (hERG, K_v_11.1) potassium channels. The blockage of these channels by pharmacological compounds is associated to drug-induced long QT syndrome (LQTS), which is a life-threatening disorder characterized by ventricular arrhythmias and defects in cardiac repolarization that can be illustrated using cardiomyocytes derived from human-induced pluripotent stem cells (hiPS-CMs). This study was meant to assess the modification in hiPS-CMs excitability and contractile properties by BeKm-1, a natural scorpion venom peptide that selectively interacts with the extracellular face of hERG, by opposition to reference compounds that act onto the intracellular face. Using an automated patch-clamp system, we compared the affinity of BeKm-1 for hERG channels with some reference compounds. We fully assessed its effects on the electrophysiological, calcium handling, and beating properties of hiPS-CMs. By delaying cardiomyocyte repolarization, the peptide induces early afterdepolarizations and reduces spontaneous action potentials, calcium transients, and contraction frequencies, therefore recapitulating several of the critical phenotype features associated with arrhythmic risk in drug-induced LQTS. BeKm-1 exemplifies an interesting reference compound in the integrated hiPS-CMs cell model for all drugs that may block the hERG channel from the outer face. Being a peptide that is easily modifiable, it will serve as an ideal molecular platform for the design of new hERG modulators displaying additional functionalities.

## 1. Introduction

The K_v_11.1 channel (hERG) and the corresponding I_Kr_ current play a major role in cardiac repolarization (Appendix A). Hence, dysfunction in hERG channels is associated with cardiac arrhythmias that can lead to sudden cardiac death. Long QT syndrome is one of the cardiac diseases that has been studied in the greatest detail. Type II long QT syndrome is associated with a decrease in I_Kr_ currents, as opposed to short QT syndrome, which is associated with increased I_Kr_ currents. Type II long QT syndrome is called “inherited” when a mutation in the gene encoding hERG, *KCNH2*, gives rise to a loss of channel function. In contrast, long QT syndrome is called “acquired” when hERG is inhibited by one of the numerous drugs whose off-target activities alter channel function. The implication of hERG in cardiac arrhythmias in such mechanistic modalities (i.e., mutations or drugs) remains a driving force to study the structure, biophysics, and pharmacology of this channel. Non-selective compounds that block hERG channels comprise antibiotics (such as erythromycin), anti-psychotics (chlorpromazine), or antihistamines (terfenadine). More selective tools to study hERG channel function include dofetilide, astemizole, and E4031, all acting on the intracellular face of the channel.

As acquired long QT is a life-threatening condition, each newly discovered drug must pass a well-defined safety screen protocol that comprises extensive testing on the hERG channel function. The gold standard for this preclinical cardiac safety testing previously included a direct electrophysiological assessment of hERG channel properties. However, the measurement of hERG block and repolarization delay is by itself an insufficient predictor of drug-induced arrhythmias. For these reasons, drug assessment also relies on a direct testing on cardiomyocytes derived from human-induced pluripotent stem cells (hiPS-CMs) within the Comprehensive in vitro Proarrhythmia Assay (CiPA) paradigm (www.cipaproject.org). The use of this protocol is now recommended, as it allows a more comprehensive evaluation of the potential cardiac liabilities and thus gives more trustworthy information to discriminate drugs that can reach the upper phases of clinical trials. The use of hiPS-CMs, in which the expression levels of ion channels as well as related signaling pathways (G protein-coupled receptors (GPCRs) and kinases) are far more relevant than in stable cell lines expressing the hERG channel, undoubtedly contributes to a better understanding and prediction of the human clinical responses.

In this report, we focused on BeKm-1, a toxin that was originally isolated from the venom of *Buthus eupeus* scorpion [1]. BeKm-1 is a 36 amino-acid peptide of 4091.7 Da molecular weight containing three disulfide bridges. According to its 3D structure, the peptide adopts an α-helix and three β-strands organized in a twisted antiparallel β-sheet [2]. By homology with other toxins, BeKm-1 is a member of the γ-KTx subfamily of scorpion toxins. The most important feature of BeKm-1 is its high selectivity and affinity for hERG channels that it blocks in the closed state through a pore-obstruction mechanism with a reported IC_50_ value of 3.3–15.3 nM [1,2,3,4]. This reported value can be compared to some of the best hERG inhibitors: E4031 (IC_50_ = 7.7 nM [5]), dofetilide (IC_50_ = 12 nM [6]), and astemizole (IC_50_ = 26 nM [7]). The proarrhythmic effect of BeKm-1 was evidenced in rabbit heart, a species expressing ERG channel, where it prolongs the QT interval in the electrocardiogram (ECG) [8]. Yet, the effect of BeKm-1 has never been tested on hiPS-CMs. Nevertheless, this investigation remains important for several reasons. First, BeKm-1 preferentially binds the hERG outer mouth as opposed to the classical model of hERG channel block (from inside the cell) [9]. While a vast majority of hERG blockers are shown to act from the inside, the mechanism of action of many other drugs are unknown. Therefore, it cannot be claimed conclusively that they all block the channel from the inside. Therefore, having a reference compound that acts from the outer face of the channel may be an interesting addition. Second, BeKm-1 blocks the closed state of the channel [4], instead of the open state classically reported for other blockers. Third, BeKm-1 is highly selective for the hERG channel in cardiomyocytes, hence theoretically sparing other ion channels. Fourth, BeKm-1 is one of the highest affinity blockers of hERG known so far, as mentioned above. Finally, this peptide can be modified to become a reporter tracer for studying channel distribution [10] or membrane expression levels [3], which is hardly feasible with other blockers.

Herein, we characterize the proarrhythmic effects of BeKm-1 on non-commercial hiPS-CMs using a variety of approaches. We conclude that a block through the outer pore does not constitute enough of a differentiating factor compared to other classical blockers in investigating the role of hERG channels. In contrast, the fact that it is a peptide will help the design of novel interesting molecular tools with added functionalities in the future.

## 2. Results

### 2.1. BeKm-1 Is a High-Affinity Blocker of hERG Channel

We first analyzed the dose–response blocking efficacy of BeKm-1 compared to classical hERG blockers, E4031 and dofetilide, on hERG currents recorded from HEK293 cells. For that purpose, we used an automated high-throughput patch-clamp system. Although it is always difficult to compare experimental IC_50_ values with those of the literature because the experimental conditions change significantly among laboratories, recordings of tens of cells for a given compound concentration with a standardized protocol and drug administration method ensures the generation of highly comparable results. As shown in Figure 1A, HEK293 cells expressed significant levels of I_Kr_ currents upon repolarization during the time course of our voltage protocol. These currents were efficiently inhibited by the hERG blockers tested. Fit of the concentration–response data for BeKm-1 inhibition experiments revealed an IC_50_ of 1.9 ± 0.3 nM (*n* = 44 cells; Figure 1B). The block of hERG currents reached a plateau of 90% at the maximally effective doses of the peptide (100 nM), indicating that BeKm-1 is not a complete pore blocker, which is coherent with earlier observations [11]. In comparison, dofetilide and E4031 also inhibited hERG currents but with higher IC_50_ values of 7.2 ± 0.9 nM (*n* = 34 cells; 3.8-fold lower) and 30.6 ± 1.5 nM (*n* = 40 cells; 16.2-fold lower), respectively (Figure 1B). The maximal block of hERG currents by these two compounds reached 99% (dofetilide at concentrations above 100 nM) and 90% (E4031 above 333 nM). These comparisons were performed in similar experimental conditions and indicate that BeKm-1 blocks with a better IC_50_ the hERG channel, even though there is agreement that the nature of the protocols may affect these absolute values [4]. For the following experiments, we used a concentration of BeKm-1 between 2 and 5 nM, a range in which significant hERG channel inhibition occurs, but that leaves a fraction of the channels untouched. We exclusively studied BeKm-1 effects, as those of E4031 and dofetilide have been extensively reported in the past.

### 2.2. BeKm-1 Produces Characteristic Modifications of the Action Potential (AP) in hiPS-CMs

The effect of an intermediate concentration of BeKm-1 (5 nM) that does not fully block hERG channels was tested on AP of spontaneously beating hiPS-CMs (Figure 2A). For this study, we used a non-commercial hiPS cell clone (C2a) that has been fully characterized earlier in the laboratory [12]. The properties of the cardiomyocytes derived from this clone may differ to some extent from those of commercial clones, but this should not affect the nature of the conclusions derived from our investigations. As shown, the administration of 5 nM BeKm-1 produced a significant reduction of AP frequency from 2.50 ± 0.20 Hz (vehicle, *n* = 7) to 0.70 ± 0.26 Hz (BeKm-1, *n* = 7) (Figure 2A,B), which is associated with an increase of rhythm variability (Figure 2C). The effects on AP characteristics also included the depolarization of the maximum diastolic potential (MDP) from −52.2 ± 1.7 mV to −36.1 ± 2.3 mV (*n* = 7, Figure 2D) and the decrease of maximal upstroke velocity from 6.9 ± 0.9 mV/s (vehicle, *n* = 7) to 4.6 ± 0.9 mV/s (BeKm-1, *n* = 7), representing an average 1.5-fold reduction (Figure 2E). Of note, AP overshoot (amplitude of the AP over 0 mV) was slightly but not significantly modified by BeKm-1 (29.9 ± 3.1 mV for vehicle vs. 22.2 ± 6.4 mV with BeKm-1, *n* = 7, Figure 2F). The action potential duration (APD) at 30% and 50% of repolarization were spared, which is consistent with the role of hERG in the late phase of repolarization in these cells (Figure 2G). In contrast, BeKm-1 was able to delay the repolarization of these cells, increasing the APD at 90% repolarization (APD_90_) from 95.0 ± 8.4 ms (vehicle, *n* = 7) to 117.4 ± 11.3 ms (BeKm-1, *n* = 7; 23% increase) (Figure 2G). While these values could theoretically be corrected for the BeKm-1-induced variability in cycle length (Figure 2C) using the Fridericia cube root formula, the corrected data would be meaningless because of the additional depolarization in MDP (Figure 2D).

One of the features associated to the prolongation of the APD in acquired LQTS is the presence of early afterdepolarizations (EADs) due to the reactivation of voltage-gated calcium channels (VGCC). In these spontaneously beating ventricular cells, EADs were never observed upon BeKm-1 application probably because of the MDP depolarization. As illustrated by a Poincaré plot, the APD_90_ variability of hiPS-CMs was also increased after treatment with 5 nM BeKm-1 (Figure 2H). Importantly, these effects were produced by BeKm-1 in a reversible fashion, which was a property that was difficult to assess in automated patch-clamp experiments due to technical issues. Wash of the toxin fully restored all parameters (Figure 2B–H; *n* = 5 for each panel). Coherently, all the parameters of the AP modified by BeKm-1 changed according to similar kinetics, suggesting that BeKm-1 affects the AP solely by acting on hERG channels (Appendix A). A representative time course of these parameters also shows a rather rapid restitution of these parameters after toxin washout (Appendix A).

Although the effect of BeKm-1 on APD_90_ was expected from a block of hERG channels, the other effects (i.e., membrane depolarization, reduction in AP frequency, and decrease of maximal upstroke velocity) are most likely a secondary consequence of the absence of I_K1_ in hiPS-CMs due to incomplete maturation level of these cells during the differentiation process (a recurrent problem in differentiating hiPS into cardiomyocytes). The 5 nM concentration of BeKm-1 was chosen to spare a fraction of the hERG channels in these cells. Therefore, we also tested a higher concentration (100 nM) in order to affect all hERG channels. As shown on the representative trace, 100 nM BeKm-1 produced a complete beating arrest by a major membrane depolarization that was sufficient to inactivate Na_v_1.5 and Ca_v_1.2 channels (Figure 2I). This observation is coherent with the fact that a stronger I_K1_ current would be needed in hiPS-CMs to maintain a more negative membrane potential typical for CMs. Hence, to get a more realistic hint of how BeKm-1 effects would be circumscribed in a more mature human cardiomyocyte, we also used dynamic clamp, which is a technique that can simulate the presence of I_K1_ in these cells. In this technique, a current density of 2.8 pA/pF is injected in order to successfully reach an RMP value of −80 mV. Those cells that did not reach this RMP value were rejected from the analyses. Representative APs, paced at a constant cycling length of 700 ms with I_K1_, are shown before and after the application of 5 nM BeKm-1 (Figure 3A,B). In these conditions, hiPS-CMs showed the occurrence of APs with EADs (Figure 3A), which are events not observed on spontaneously beating cardiomyocytes. Logically, these APs were excluded from further analysis of AP parameters. I_K1_ in silico injection efficiently allowed for setting a real resting membrane potential (RMP) at −80 mV in these hiPS-CMs, causing the cessation of spontaneous APs (not shown). As a result of I_K1_ taking the relay over I_Kr_, there was no effect of 5 nM BeKm-1 on RMP (Figure 3C). Consistent with the fact that depolarizing channels can fully recover at these transmembrane potentials, no significant effect was observed on the overshoot (Figure 3D), nor on the maximal upstroke velocity (Figure 3E). As expected from a late implication of hERG channels in the AP, and in coherence with experiments on spontaneous APs, BeKm-1 did not alter the APD_30_ and APD_50_, but it significantly prolonged the APD_90_ from 121.9 ± 11.0 ms (vehicle, *n* = 10) to 156.7 ± 16.9 ms (BeKm-1, *n* = 10; 29% prolongation on average), which were all measured at a constant cycle length and MDP (Figure 3B,F). In addition, cells dynamically clamped with I_K1_ also showed greater APD_90_ variability after treatment with 5 nM BeKm-1 (Figure 3G). The average prolongation of APD_90_ and variability is also detailed cell by cell (Appendix A).

### 2.3. Proarrhythmic Effects of BeKm-1 in Dynamic Clamp Conditions

Prolongation of the repolarization phase allows VGCC and voltage-gated sodium channels to recover from inactivation and reactivate, further depolarizing the membrane and generating EADs. Drug-induced LQTS has been regularly associated with frequent EADs, sometimes leading to torsades de pointes (TdP). Consistent with these observations, most of the cells showed occurrences of EADs after treatment with 5 nM BeKm-1, whereas none of these cells showed any type of arrhythmogenic feature in vehicle conditions (Figure 3A,H,I). These results highlight the ability of BeKm-1 to recapitulate drug-induced LQTS electrophysiological features in dynamically clamped hiPS-CMs where the RMP stays at −80 mV.

### 2.4. BeKm-1 Alters Excitatory Field Potentials of hiPS-CMs

The excitability abnormalities recorded on single hiPS-CM APs should translate into larger abnormalities when these cells are interacting with each other in monolayers. To assess this issue, we investigated the electrical excitability of hiPS-CM monolayers using the CardioExcyte96 system (Nanion). This system monitors both the electric field potential (EFP), which reflects the concerted electrical activity of the cell monolayer, and the impedance changes due to cell contractions, which will be described below. With regard to the EFP properties, application of 2 nM BeKm-1, again an intermediate concentration close to the IC_50_ value, had a dual effect: a slight increase in EFP frequency (0.13 ± 0.01 Hz for vehicle vs. 0.15 ± 0.01 Hz for BeKm-1, *n* = 6) (Figure 4A,B), and a slight increase in the duration of the EFP (2.8 ± 0.1 s for vehicle vs. 3.4 ± 0.2 s with BeKm-1, *n* = 5), which is coherent with the observed increase in APD_90_ (Figure 4C,D). The increase of EFP duration and the presence of spontaneous depolarization (Figure 4A) is coherent with the proarrhythmic effect of the peptide observed in AP recordings (Figure 3I). As saturating concentrations of BeKm-1 (100 nM) induced the arrest of spontaneous APs in individual cells, we were interested to see what may happen on a monolayer of hiPS-CMs. Using 100 nM BeKm-1, we observed a significant increase in EFP frequency (0.2 ± 0.1 Hz for vehicle vs. 0.7 ± 0.1 Hz with BeKm-1, *n* = 5, Figure 4E,F), sustained waveforms of tachyarrhythmia, and TdP-like polymorphic waveforms (Figure 4E).

### 2.5. BeKm-1 Indirectly Alters the Ca^2+^ Dynamics and Contraction Properties of hiPS-CMs

As Ca^2+^ influx abnormalities are likely to be at the root of electrophysiological disorders observed in the presence of BeKm-1, we sought to investigate the effects of this compound on hiPS-CM calcium dynamics. In order to test this, intracellular Ca^2+^ transients (CaT) were monitored by confocal microscopy using the Fluo4 Ca^2+^-sensitive dye in presence of 5 nM BeKm-1. As shown in Appendix A, on spontaneously beating cardiomyocytes, BeKm-1 profoundly altered the shape and frequency of CaT. As shown upon quantification, a great variability was introduced in the intervals of CaT (Appendix A). The average interval was raised from 3.3 ± 1.7 sec (*n* = 8, vehicle) to 4.9 ± 2.2 sec (*n* = 8, 5 nM BeKm-1) (Appendix A). In addition, we observed a reduction of CaT amplitude from 1.8 ± 0.2 (*n* = 8, vehicle) to 0.8 ± 0.2 (*n* = 8, 5 nM BeKm-1) (Appendix A). Integration of the Ca^2+^ signals indicates a non-significant decrease of the cytoplasmic Ca^2+^ handling (Appendix A). The altered shape of the CaT was due to a slower time to peak (1.1 ± 0.1 s with vehicle vs. 2.2 ± 0.4 s with 5 nM BeKm-1, *n* = 8) (Appendix A), possibly indicating a slower release from internal stores associated to a reduced entry by VGCC (largely due to the impact of RMP depolarization). However, no alterations were observed in the kinetics of the decay phase (Appendix A), which is coherent with preserved recapture properties by sarco/endoplasmic reticulum Ca^2+^ ATPase (SERCA) pumps and extrusion by sodium/calcium exchangers (NCX). These results indicate that the cell surface electrical effects of BeKm-1 may potentially also impact the calcium dynamics of hiPS-CMs.

The results described above on the excitability alterations of hiPS-CMs indicate that BeKm-1 should also impair excitation–contraction coupling (ECC) in these cells. In spontaneously beating cardiomyocytes, the toxin should impact the contractility in a coherent manner with spontaneous APs and CaT. We investigated the contractions using the CardioExcyte 96 system. A superposition of representative traces before and after the application of 2 nM BeKm-1 shows that the peptide had little impact on the amplitude of the contractions, but it altered the kinetics of the rising and decay phases (Appendix A). The average contraction amplitude was not significantly different from the control, which was probably due to the limits of this system for measuring variations in contractile forces (Appendix A). In contrast, we observed an increase in the duration of the rising phase of contraction from 203.5 ± 11.3 ms (vehicle, *n* = 8) to 262.6 ± 16.9 ms (BeKm-1, *n* = 8, 1.29-fold slowing) (Appendix A) and an increase of the duration of the relaxation phase from 719.2 ± 45.8 ms (Vehicle, *n* = 8) to 862.1 ± 82.0 ms (BeKm-1, *n* = 8; 19.8% slowing) (Appendix A). However, none of the pulse widths (PW) PW_30_, PW_70_, and PW_90_ parameters were significantly altered (Appendix A). However, there was a significant alteration of the PW_95_ parameter, which is coherent with the shape of the contraction signal (Appendix A). More interestingly, we investigated the proarrhythmic effects of BeKm-1 on the contractility of hiPS-CMs. The interbeat interval and its variability were significantly increased by 2 nM BeKm-1 according to a Poincaré plot analysis of this parameter (Appendix A). The extent of interbeat interval increase was 16.7% raising from 2.03 ± 0.1 s (vehicle, *n* = 8) to 2.37 ± 0.2 s (BeKm-1, *n* = 8) (Appendix A). Coherent with the proarrhythmic properties of 2 nM of BeKm-1, both the mean SD1, the standard deviation of instantaneous contraction-to-contraction interval variability, and SD2, the continuous long-term interval variability, increased by 2- and 1.7-fold, respectively (Appendix A).

## 3. Discussion

In this study, we describe the effects of the high-affinity hERG blocker BeKm-1 on hiPS-CMs. By using a variety of approaches, we efficiently assessed these effects on the electrophysiological, calcium handling, and contractile properties of hiPS-CMs providing a large response picture of this natural peptide. The affinity of 1.9 nM of BeKm-1 for hERG channels, as determined in an automated patch-clamp study, was similar if not better to that previously described in other studies [1]. The effects of a mid-dose application of BeKm-1 on spontaneously beating hiPS-CMs consistently included cellular repolarization delay, negative chronotropic effects on APs, CaT, and monolayer contractions, which were accompanied with the higher rhythm variability of these parameters. This is in agreement with other studies investigating the effects of BeKm-1 on more integrated models [8]. Of note, negative chronotropic response is also in accordance with studies with common hERG blockers E4031 and dofetilide on hiPS-CMs [13] and on rabbit isolated hearts [14]. The bradycardic effects described here for BeKm-1 have already been reported for E4031 and dofetilide on integrated models [15]. They are most likely due to the expression of hERG channels in the sinoatrial node (SAN), in which I_Kr_ plays a major role in pacemaker activity [16]. Interestingly, the MDP in these cells is about similar as MDP observed in our hiPS-CMs. Obviously, the elevation of MDP in SAN cells and in hiPS-CMs reduces the availability of depolarizing channels such as VGCCs, which should account for the effects observed on our spontaneous APs, and the alterations in contractile properties and AP, CaT, and beating frequencies.

We reported on the proarrhythmic effects of BeKm-1 both at the cellular level (EADs) and in cardiomyocytes layers with increased EFP duration and waveforms of tachyarrhythmia. Similar results have been reported using high doses of E4031 in order to develop an in vitro model of TdP [17]. We found that high concentrations of BeKm-1 arrested the automaticity of hiPS-CMs, similarly to what was observed in previous studies on isolated SAN cells [16]. In pacemaker cells, the arrest of pacemaker activity following I_Kr_ blockade is attributed to the absence of connection with atrial tissue [18]. In the intact heart, the expression of I_K1_ in the atria exerts a hyperpolarizing load on the SAN. This electrotonic load controls SAN repolarization under conditions of I_Kr_ blockade. The arrest of automaticity in hiPS-CMs can be attributed to the poor expression of I_K1_ in these myocytes, underscoring the necessity to reproduce I_K1_ in silico for pharmacologic testing of I_Kr_ inhibition. Notably, this is also part of the CiPA initiative to address limitations regarding the electrophysiological immaturity of hiPS-CMs (i.e., absence of the I_K1_ setting RMP to −80 mV) that seeks to combine in vitro drug testing and computational modeling of cardiac currents during AP [19]. To that extent, in silico I_K1_ injection allowed us to better model the effects of BeKm-1 on ventricular APs and obtain more insight into its arrhythmic properties. In the future, the use of more mature cardiomyocytes, possibly from commercial sources as well, may help circumvent some of the drawbacks associated with the poor expression of I_K1_. As attested by the presence of AP triangulation (slowing of late phase repolarization) and frequent EADs, the effects of BeKm-1 are consistent with what is observed with other hERG blockers on ventricular cardiomyocytes [20]. APD prolongation and EADs are at the basis of the QT interval prolongation and the arrhythmicity (sometimes with the occurrence of TdP) observed in drug-induced LQTS, demonstrating that BeKm-1 can induce pathological conditions in hiPS-CMs.

## 4. Materials and Methods

### 4.1. Materials

BeKm-1 was purchased from Smartox Biotechnology (Saint-Egrève, France).

### 4.2. Human-Induced Pluripotent Stem Cell-Derived Cardiomyocytes and HEK-293 Cell Culture

*hiPS-CMs*: All experiments were performed on the human-induced pluripotent stem cell (hiPSC) clone C2a that has been previously characterized, including cardiac differentiation and electrophysiology data [12]. Cells were expanded on stem cell-qualified matrigel-coated plates (0.05 mg/mL; BD Bioscience, Le Pont de Claix, France) and cultured in StemMACS iPS-BREW XF medium (Miltenyi Biotec, Paris, France). Cells were passaged every 45 days using Gentle Cell Dissociation Buffer (StemCell Technologies, Grenoble, France). Prior to differentiation, cells were cultured as a monolayer in StemMACS iPS-brew supplemented with 1 × Y-27632 Rho-associated, coiled-coil containing protein kinase (ROCK) inhibitor (StemCell Technologies, Grenoble, France). When hiPSCs reached proper confluence, they were differentiated into cardiomyocytes using previously described « Matrix sandwich » protocol [21,22]. At day 20 ± 1 of differentiation, beating cardiomyocytes were manually dissected and isolated from non-beating cells. Briefly, cardiomyocytes were washed with phosphate-buffered saline (PBS) and treated with TrypLE Express (10x) (Thermo Fisher Scientific, Illkirch-Graffenstaden, France) for 5 min at 37 °C. The enzyme was inactivated by adding RPMI1640 medium containing 20% (vol/vol) KnockOut Serum Replacement (KSR) (Thermo Fisher Scientific, Illkirch-Graffenstaden, France). The supernatant was removed, and cells were suspended in RPMI1640 medium containing 20% (vol/vol) KSR and ROCK inhibitor. Then, the cells were plated on specific supports suitable for functional assessments.

*HEK293 cells*: HEK-293 cells stably expressing the human hERG channel (from Bioprojet, Rennes, France) were cultured in Dulbecco’s Modified Eagle’s Medium (DMEM) supplemented with 10% fetal calf serum, 4.5 g/L glucose, 2 mM glutamine, 10 U/mL penicillin, and 10 μg/mL streptomycin (Gibco, Grand Island, NY, USA), 400 µg/mL G418 (Thermo Fisher Scientific, Illkirch-Graffenstaden, France), and incubated at 37 °C in a 5% CO_2_ atmosphere. For electrophysiological recordings, cells were detached with accutase, and floating single cells were diluted (≈300,000 cells/mL) in medium containing (in mM): 4 KCl, 140 NaCl, 5 glucose, and 10 4-(2-hydroxyethyl)-1-piperazineethanesulfonic acid (HEPES) (pH 7.4, osmolarity 290 mOsm).

### 4.3. Manual and Automated Patch-Clamp Experiments

Manual patch-clamp experiments were performed on single-cell dissociated hiPS-CMs for the recording of APs in the current clamp and in the dynamic clamp mode, while an automated patch clamp was used to record from HEK293 cells stably expressing hERG channels.

*Manual patch clamp*: Single-cell dissociated hiPS-CMs were plated at low density on matrigel-coated 35 mm Petri dishes (Nunc from Thermo Fisher Scientific, Illkirch-Graffenstaden, France). Electrophysiological experiments were performed 12 to 14 days after dissociation. APs were recorded in the current-clamp mode, using the amphotericin-B-perforated patch-clamp technique. All experiments were conducted at 37 °C. APs were recorded either at basal state (spontaneous APs) or using the dynamic patch-clamp method to electronically mimic the expression of the inward rectifier potassium current, I_K1_ [23,24]. The resting membrane potential (RMP) was set to −80 ± 5 mV by injecting 2 to 2.8 pA/pF in silico I_K1_ density. This level of current was set so that about 80% of the ventricular-like hiPS-CMs could reach these physiological RMP values. The remaining cells were discarded from the recording process. For cells that successfully reached RMP of −80 mV, depolarization was produced every 700 ms with a 3035– pA/pF current density pulse in these dynamic patch-clamp conditions. Both cell pacing and I_K1_ injection were driven by custom-made software and a National Instrument A/D converter (NI PCI-6221, Austin, Texas, USA) connected to the current command of the amplifier. Cells were perfused with extracellular solution containing (in mM): 140 NaCl, 4 KCl, 0.5 MgCl_2_, 1 CaCl_2_, 10 HEPES, 10 glucose; pH 7.4 (NaOH). Low resistance borosilicate glass pipettes (23– MΩ; Sutter Instruments, Novato, California, USA) were filled with a solution containing (in mM): 125 K-gluconate, 5 NaCl, 20 KCl, 5 HEPES; pH 7.2 (KOH), supplemented with 0.22 mg/mL amphotericin-B. Stimulation and data recordings were performed using an Axopatch 200B amplifier controlled by Axon pClamp 10.6 software through an A/D converter (Digidata 1440A, Molecular Devices, San José, California, USA). The data were analyzed using the Clampfit 10.6 software. APs were classified as nodal-, atrial-, and ventricular-like based on their duration, maximum upstroke velocity (dV/dtmax), amplitude (mV), and maximum diastolic potential, as previously defined (Appendix A) [21]. For further analyses, only APs from ventricular-like cardiomyocytes were investigated, thus excluding atrial- and nodal-like cells. For further analysis, only APs from ventricular-like cardiomyocytes were investigated as determined by AP shape, thus excluding atrial-like and nodal like cells. Various AP characteristics were examined, including maximum diastolic potential (MDP), AP overshoot (i.e., amplitude of the AP over 0 mV), maximal upstroke velocity (dV/dt_max_), and AP duration at different times of repolarization (APD). For each analysis, parameters from seven consecutive APs were averaged to free results from intracellular AP variability.

*Automated patch clamp*: BeKm-1 and other hERG-acting compounds were investigated on HEK293 cells stably expressing the hERG channel using the automated patch-clamp system SyncroPatch 384PE (Nanion Technologies, Munich, Germany). Chips with single-hole and high-resistance 5.14 ± 0.02 MΩ (*n* = 384)) were used for HEK293 cell recordings. Voltage pulses and whole-cell recordings were achieved using the PatchControl384 v1.5.3 software (Nanion Technologies, Munich, Germany) and the Biomek v1.0 interface (Beckman Coulter, Villepinte, France). Prior to recordings, dissociated cells were shaken at 60 rpm in a cell hotel reservoir at 10 °C. After cell catching, sealing, whole-cell formation, liquid application, recording, and data acquisition were all performed sequentially and automatically. The intracellular solution contained (in mM): 10 KCl, 110 KF, 10 NaCl, 1 MgCl_2_, 1 CaCl_2_, 10 Ethylene Glycol Tetraacetic Acid (EGTA) and 10 HEPES (pH 7.2, osmolarity 280 mOsm), and the extracellular solution (in mM): 140 NaCl, 4 KCl, 2 CaCl_2_, 1 MgCl_2_, 5 glucose, and 10 HEPES (pH 7.4, osmolarity 298 mOsm). Whole-cell experiments were done at −80 mV holding potential, while currents triggered at −40 mV test potential were sampled at 10 kHz (sweeps every 8 s). Compounds were prepared at various concentrations in the extracellular solution supplemented with 0.3% bovine serum albumin (BSA) and distributed in 384-well plates according to a pre-designed template. Compound solutions were diluted three times in the patch-clamp recording well by adding 30 to 60 μL external solution to reach the final reported concentration and the test volume of 90 μL. The percentage of current inhibition was measured after a 13-min application time. A single concentration of peptide was tested on each cell for building the full-inhibition curves.

### 4.4. Beating Assessment and Electrical Field Potential Measurements of hiPS-CMs

The beating properties and the electrical field potentials of hiPS-CMs were evaluated with a CardioExcyte96 system. After dissociation, hiPS-CMs were seeded on NSP-96 Sensor Plates (Nanion Technologies, Munich, Germany) at a density of 40,000 viable cells per well. Sensor plates were previously coated with 10 µg/mL fibronectin for 1 h at 37 °C. The cells were maintained at 37 °C, and the medium was replaced every 2 days for 1 week. On the day of beating assessment, the whole medium was replaced 4 h prior to experiments, and cells were incubated at 37 °C onto the CardioExcyte96 system. Meanwhile, cells were monitored to ensure the full stabilization of beating parameters until BeKm-1 was tested at various concentrations (2 and 100 mM). T_fall_ and T_rise_ were determined as the time between 10% and 90% of contraction or relaxation, respectively. Pulse width (PWx) was determined as the time corresponding to the width of the signal at 90% of relaxation. Electric field potentials (EFPs) were recorded using CardioExcyte96 system. For both contraction and EFPs, 10 sequences of 30 s of acquisition were analyzed before and after BeKm-1 addition. The CardioExcyte system is best to analyze the kinetic parameters of the contractile properties of hiPS-CMs but has intrinsic limitations to assess properly variations in contractile force.

### 4.5. Calcium Transient Recordings

Calcium transients of hiPS-CMs were evaluated using the Fluo-4 reported dye and a Nikon A1R confocal microscope. Dissociated hiPS-CMs were plated and maintained on µ-slide 8-well IBIDI plates for 12 ± 1 days prior to experiments. On the day of the experiments, cells were loaded for 20 min with 3 µM Fluo-4 AM Ca^2+^ indicator (Thermo Fisher Scientific, Illkirch-Graffenstaden, France) in culture medium at 37 °C. Then, to allow complete de-esterification of the Ca^2+^ probe, the medium was replaced and cells were re-incubated at 37 °C for another 20 min. All recordings were performed in Tyrode’s solution, cells being placed at 37 °C in an experimental chamber. Confocal microscopy was performed using a Nikon A1R confocal inverted microscope (Nikon Corp, Tokyo, Japan) with a Nikon X60 Plan-Apo numerical aperture 1.4 oil-immersion objective (MicroPICell core facility). Fluo-4 AM was excited using the 488 nm line from an argon laser and fluorescence emission was measured at >505 nm. Images were acquired between 15.5 and 56.4 frames/s in X-Y mode, with a resolution of 1024 × 1024 pixels (9.5 µm per pixel).

### 4.6. Statistics

Data are represented as mean  ±  SEM. Significance was calculated using a two-sided paired Student’s t-test for two group comparisons or a Fisher test for percentages comparison, as indicated in the figure legends and considered as significant at *p* < 0.05. Analysis was done using GraphPad Prism V6 (GraphPad, San diego, CA, USA). 

## 5. Conclusions

BeKm-1 is a compound of interest that may be considered later, with further extended investigation, for inclusion in the list of positive reference controls for drug arrhythmic testing within the CiPA protocol. This conclusion is also lead by the remarkable properties of BeKm-1. First, the peptide showed better IC_50_ values than the reference compounds E4031 and dofetilide tested in this study (IC_50_ values of 31 nM and 7 nM, respectively). Since these assays were conducted in similar experimental conditions, we believe that BeKm-1 can be considered as one of the most potent hERG blockers reported so far. The values we report for BeKm-1 were still better than earlier reports, but this difference can be accounted from the use of an automated patch-clamp system, well-controlled experimental conditions, and an elevated number of high-quality cells recorded. Second, the peptide is favorably selective for hERG channels, and no other targets have been reported for this peptide [1]. Third, concerning its activity on I_Kr_, BeKm-1 has been widely studied on numerous models from different species so far, so that its effects are among the most well-characterized on hERG channels. Importantly, docking of the peptide has been investigated shedding lights on its mechanistic blocking properties [10]. Finally, contrary to small molecule drugs that need to cross the plasma membrane to block hERG from the cytosolic part, BeKm-1 acts from the outer mouth to block the closed-state of the channel. For that matter, BeKm-1 provides more diversity into tools to investigate the proarrhythmic characteristics of hERG blockers. Importantly, the peptide nature of BeKm-1 allows interesting chemical modifications for the grafting of new functionalities: something that is hardly feasible with classical small compound blockers. Already, engineered BeKm-1 peptides have been designed, such as fluorescent analogues [10], which will reveal themselves useful for studying hERG channel distribution or performing biochemical assays. Soon, we may envision the production of new BeKm-1 analogues such as irreversible or photosensitive blockers that should open new perspectives in the investigation of the role of hERG in human cardiomyocytes.

## Figures and Tables

**Figure 1 ijms-21-07167-f001:**
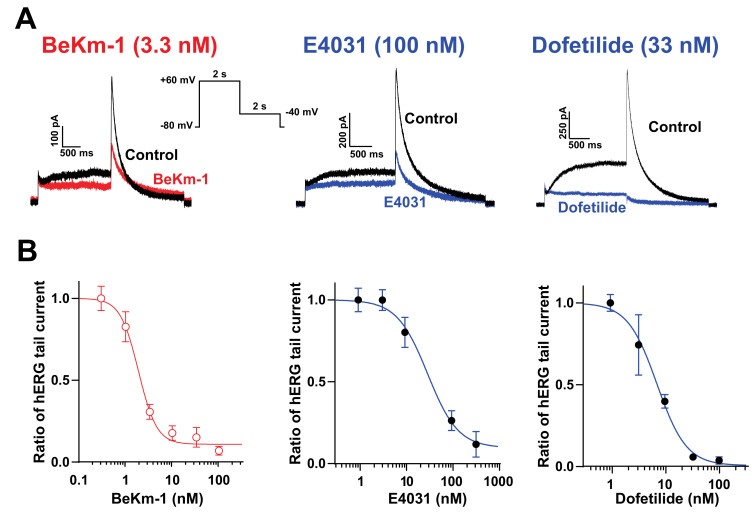
Block of hERG channels by BeKm-1 and comparison with other hERG high-affinity blockers in HEK293 cells overexpressing hERG channels. (**A**) Voltage protocol for the stimulation of hERG channels and superimposed raw traces for I_Kr_ currents in control conditions (black) or following treatment with hERG inhibitors (colored). The concentrations used for each drug are meant to show that the current block exceeds 50%. (**B**) Dose–response curve of hERG current inhibition by BeKm-1, dofetilide, and E4031. Data were fitted by a Hill equation yielding IC_50_ values of 1.9 nM (BeKm-1, *n* = 44 cells), 7.2 nM (dofetilide, *n* = 34 cells) and 30.6 nM (E4031, *n* = 40 cells), and Hill numbers of 1.9 (BeKm-1), 1.6 (dofetilide) and 1.3 (E4031).

**Figure 2 ijms-21-07167-f002:**
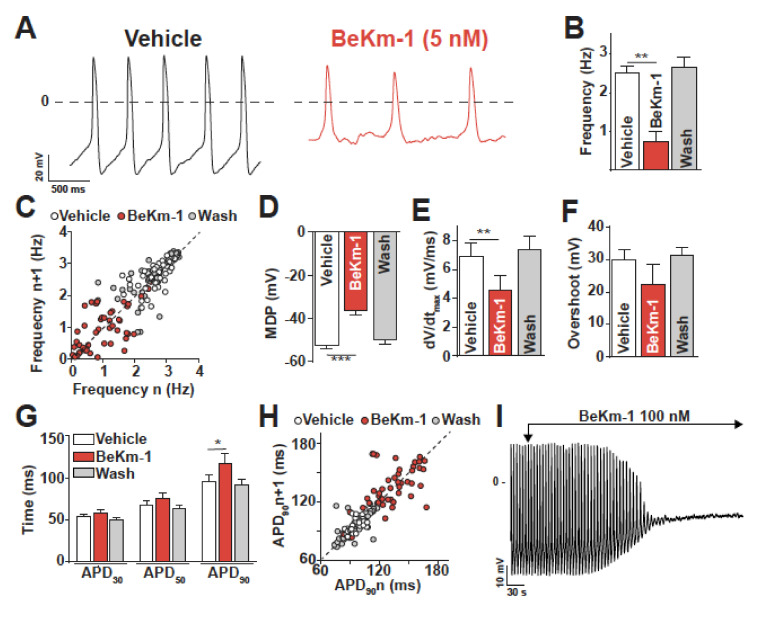
BeKm-1 alters spontaneous AP shape and frequency in human-induced pluripotent stem cell (hiPS-CMs). (**A**) Expanded traces from manual patch-clamp recordings showing the impact of 5 nM BeKm-1 on APs. (**B**) Average effect of 5 nM BeKm-1 on AP frequency. (**C**) Poincaré plot showing changes in AP frequency variability from consecutive APs for vehicle, 5 nM BeKm-1, and wash conditions. (**D**–**F**) Average effects of 5 nM BeKm-1 and reversibility of these effects on maximum diastolic potential (MDP) (D), maximal upstroke velocity (dV/dt_max_) (E) and overshoot (F). (**G**) Effect of 5 nM BeKm-1 and reversibility of this effect on action potential durations (APDs) at 30%, 50% and 90% of repolarization. (**H**) Poincaré plot showing APD_90_ variability from consecutive APs in vehicle, 5 nM BeKm-1, and wash conditions. Note the presence of red BeKm-1-associated dots underneath the cluster of white vehicles representing dots. (**I**) Representative trace showing AP arrest on a cell treated with 100 nM BeKm-1. All parameters were analyzed on ventricular-like cardiomyocytes only, as determined by objective AP shape characteristics. Thus, atrial-like and nodal-like cells were excluded from the analysis. Vehicle *n* = 7; 5 nM BeKm-1 *n* = 7; wash *n* = 5. Paired t-test: *, *p* < 0.05; **, *p* < 0.01.

**Figure 3 ijms-21-07167-f003:**
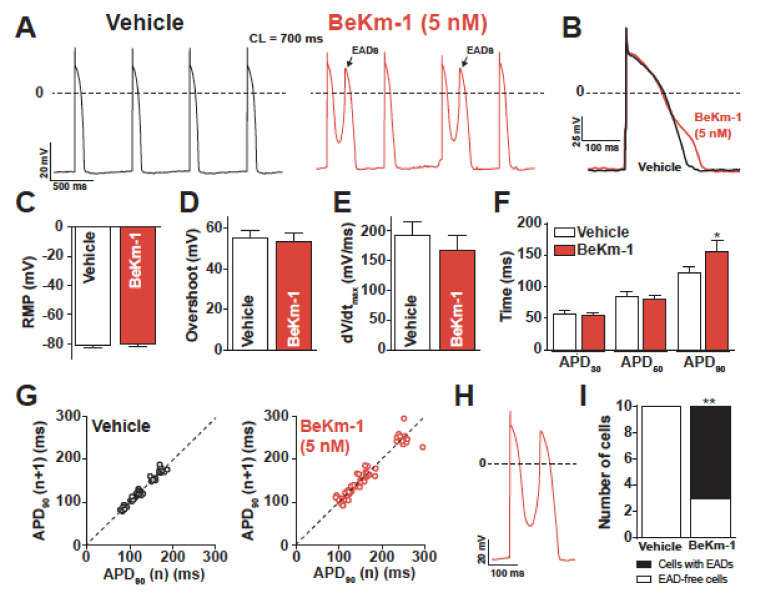
BeKm-1 mainly affects the late phase of repolarization in dynamically clamped hiPS-CMs and induces cellular arrhythmic events. (**A**) Representative APs paced at 1.4 Hz with I_K1_ before (black trace) and with 5 nM BeKm-1 application (red trace). Note the presence of early afterdepolarizations (EADs) in BeKm-1 treated cells (black arrows). (**B**) Superimposed APs raw traces of a ventricular-like cardiomyocyte before and after application of 5 nM BeKm-1 showing delayed repolarization. (**C**–**F**) Average effects of 5 nM BeKm-1 on RMP (C), overshoot (D), maximal upstroke velocity (E), and APDs at 30%, 50%, and 90% of repolarization (F). (**G**) Poincaré plots showing APD_90_ variability from consecutive APs before (left panel) and after (right panel) treatment with 5 nM BeKm-1. (**H**) Expanded representative AP showing the occurrence of an EAD during application of 5 nM BeKm-1. (**I**) Number of cells displaying EADs in both conditions. *n* = 14 cells for all panels. Paired t-tests with *, *p* < 0.05; **, *p* < 0.01. Fisher test.

**Figure 4 ijms-21-07167-f004:**
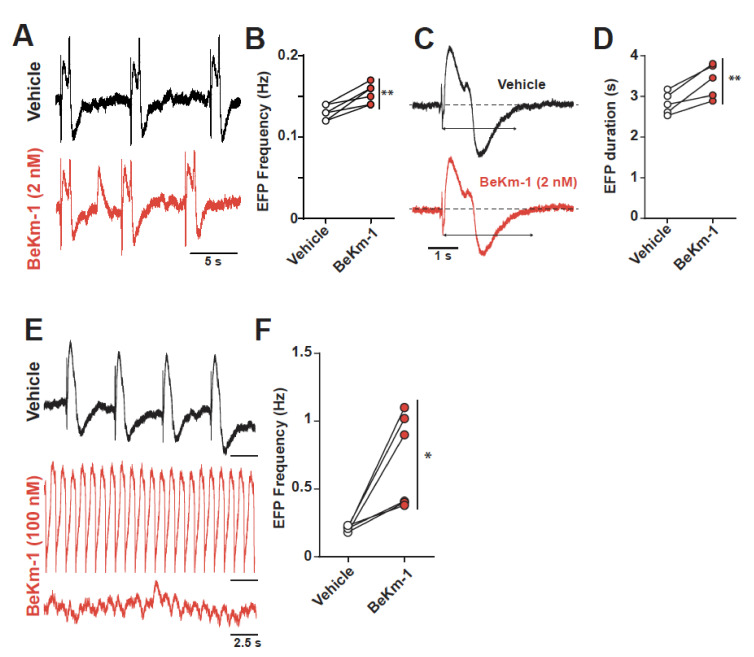
Concentration-dependent impact of BeKm-1 on electric field potential (EFP) properties of a hiPS-CMs monolayer. (**A**) Raw traces of representative EFPs showing a decrease of the frequency following a 2 nM BeKm-1 application. (**B**) Frequency quantification before and after treatment with 2 nM BeKm-1. (**C**) Expanded representative EFP showing prolonged duration following treatment with 2 nM BeKm-1. (**D**) Quantification of EFP duration before and after 2 nM BeKm-1 application. *n* = 5 independent hiPS-CMs monolayers. Paired t-tests with **, *p* < 0.01. (**E**) Raw traces of representative EFPs before and after 100 nM BeKm-1 application showing increase in beats frequency and fibrillation-like pattern. (**F**) Quantification of EFP frequency before and after treatment with 100 nM BeKm-1. *n* = 6 independent hiPS-CMs monolayers (5 sweeps of 30 s for each). Paired t-tests with *, *p* < 0.05.

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
