# Peer review of "Functional Impact of BeKm-1, a High-Affinity hERG Blocker, on Cardiomyocytes Derived from Human-Induced Pluripotent Stem Cells"

_ijms, 2020, doi:10.3390/ijms21197167_

Round 1

Reviewer 1 Report

A brief summary: De Waard et al. present a clear and easy-to-follow story about the effects of BeKm-1, a high affinity hERG blocker, on electrophysiology, calcium dynamics and contractile properties of hiPSC-CMs. The authors use elegantly both manual and automated patch-clamp methods to conduct the electrophysiological experiments. Especially, the use of dynamic patch-clamp is a clever way to investigate BeKm-1 effects in a more mature cardiomyocyte phenotype, which the hiPSC-CMs do not possess. The manuscript is also well written, except for a few weird choices of words, a couple of vague/long sentences, and some improper use of terminology.

Broad comments: I have one broad comment regarding the study. The authors report at length about the impact of BeKm-1 on calcium dynamics and contractile properties (roughly 2/7 of the text in the Results section, and 2/6 of the figures). Although interesting, all those effects are secondary to the altered electrophysiology. So, I fail to see the relevance/significance of sections “2.5. BeKm-1 alters Ca2+ handling properties of hiPS-CMs” and “2.6. The contraction properties of hiPSC-CMs are also altered by BeKm-1”. I don’t mean to underestimate the effort and expertise required for conducting those parts of the study. However, scientifically, those results are not of great interest, because BeKm-1 does not directly modulate the components of either calcium cycling machinery or contractile apparatus. Those two sections of results and the corresponding two figures should be presented as supplementary information.

Specific comments: Regarding the style of writing and choice of words:

  1. Line 51: Here and throughout the manuscript, please, use either the version “E4031” or “E-4031” not both.
  2. Line 51: I would consider starting a new paragraph from the sentence “As acquired long QT is a life-threatening…”. In its current form the paragraph is exhaustively long.
  3. Line 54: Maybe write “previously” instead of “initially”?
  4. Line 85: I would consider starting a new paragraph from the sentence “Herein, we now described the proarrhythmic…”.
  5. Lines 85-88: This is a looong sentence: “Herein, we now described the proarrhythmic effects of BeKm-1 on non-commercial hiPS-CMs using a variety of approaches and conclude that a block through the outer pore does not constitute enough a differentiating factor compared to other classical blocker in investigating the role of hERG channels.” Please, consider rewriting it for example like this: “Herein, we characterize the proarrhythmic effects of BeKm-1 on non-commercial hiPS-CMs using a variety of approaches. We conclude that a block through the outer pore does not constitute enough a differentiating factor compared to other classical blocker in investigating the role of hERG channels.”
  6. Lines 105-106: Think “lower” should be “higher”, right? Or am I missing something here?
  7. Line 162: Why “again”? To what is this “again” referring to?
  8. Line 170: I would consider starting a new paragraph from the sentence “Although the effect of BeKm-1 on APD90 was expected…”.
  9. Lines 178-179: The following sentence is written kind of backwards, in my opinion: “This observation is coherent with the fact that IK1 current is required to maintain the membrane potential in hiPS-CMs.” Maybe it could be rewritten like this: “This observation is coherent with the fact that a stronger IK1 current would be needed in hiPS-CMs to maintain a more negative membrane potential typical for CMs.”
  10. Line 181: I would write “can simulate” instead of “mimics”.
  11. Line 184: “AP” instead of “Ap”.
  12. Lines 188-189: Second half of the sentence is a bit cumbersome. I would write: “…at -80 mV in these hiPS-CMs, causing the cessation of spontaneous APs (not shown).”
  13. Lines 232-234: Weird sentence structure. I would write instead: “As saturating concentrations of BeKm-1 (100 nM) induced arrest of spontaneous APs in individual cells, we were…”
  14. Line 247: In my opinion the subheading is misleading. That is, BeKm-1 does not alter Ca2+ handling Instead, it affects indirectly the intracellular Ca2+ dynamics. So, a proper subheading would be something like “BeKm-1 alters the Ca2+ dynamics in hiPS-CMs”.
  15. Line 257: Weird choice of words: “decreases Ca2+ handling in the cytoplasm”. Write instead, for example, “Ca2+ transient amplitude in the cytoplasm”. On a related note, please, consider introducing the abbreviation “CaT” or “CT” for “Ca2+ transient” and use it throughout the manuscript.
  16. Lines 263-264: Same comment here as on line 247. BeKm-1 does not alter Ca2+ handling. Instead, it affects indirectly the intracellular Ca2+ dynamics.
  17. Line 295: What is the meaning of the “*” sign in Figure 6E?
  18. Line 326: I don’t understand, what “reduction of ECC” means. Please, consider rewriting that.
  19. Line 326: “Ca2+ transient” instead of “transient”.
  20. Line 327: I would start a new paragraph from “We reported proarrhythmic effects…”. That way the Discussion section would not be one single, exhaustively long paragraph.
  21. Line 348: I’m not sure, what the authors mean to say here: “…demonstrating that BeKm-1 may interestingly recapitulate some pathological properties of hiPS-CMs.” Maybe you could write instead: ““…demonstrating that BeKm-1 can induce pathological conditions in hiPS-CMs.”
  22. Lines 385-390: This is a quite vague description: “Between 2 and 2.8 pA/pF in silico IK1 density was generally sufficient to bring resting membrane potential (RMP) to -80 ± 5 mV. Cells whose RMP didn’t reach these physiological ventricular values were excluded from the recording process. During dynamic patch-clamp, cells were depolarized every 700 ms with a 30-35 pA/pF current density pulse. This current density was set in such a manner that not all ventricular cardiomyocytes were able to reach an RMP value of -80 mV.” Could you be a bit more specific/quantitative about your approach? Also, I guess “ventricular cardiomyocytes” should be “ventricular-like hiPS-CMs”, right?
  23. Line 473: Weird choice of words. I would write “contrary to” instead of “at odd with”.

Author Response

Reviewer 1

A brief summary: De Waard et al. present a clear and easy-to-follow story about the effects of BeKm-1, a high affinity hERG blocker, on electrophysiology, calcium dynamics and contractile properties of hiPSC-CMs. The authors use elegantly both manual and automated patch-clamp methods to conduct the electrophysiological experiments. Especially, the use of dynamic patch-clamp is a clever way to investigate BeKm-1 effects in a more mature cardiomyocyte phenotype, which the hiPSC-CMs do not possess. The manuscript is also well written, except for a few weird choices of words, a couple of vague/long sentences, and some improper use of terminology.

Broad comments: I have one broad comment regarding the study. The authors report at length about the impact of BeKm-1 on calcium dynamics and contractile properties (roughly 2/7 of the text in the Results section, and 2/6 of the figures). Although interesting, all those effects are secondary to the altered electrophysiology. So, I fail to see the relevance/significance of sections “2.5. BeKm-1 alters Ca2+ handling properties of hiPS-CMs” and “2.6. The contraction properties of hiPSC-CMs are also altered by BeKm-1”. I don’t mean to underestimate the effort and expertise required for conducting those parts of the study. However, scientifically, those results are not of great interest, because BeKm-1 does not directly modulate the components of either calcium cycling machinery or contractile apparatus. Those two sections of results and the corresponding two figures should be presented as supplementary information.

Response: We thank the reviewer for this clever appreciation of the manuscript. We do agree that these cardiomyocytes are in part immature (but this is a general issue when working with cardiomyocytes derived from human iPS). Hence, indeed, because of this level of immaturity, the BeKm-1 effects we report are largely secondary effects of the electrophysiological alterations. We see no problem at transferring these findings into the supplementary information. They remain important for us (at the technical level) because we hope to be able soon to work on more mature cardiomyocytes, which will allow a refined evaluation of the effects of BeKm-1 analogues (to come) on these parameters. The two paragraphs were merged to make them appear shorter and the figures and figure legends have been transferred to the supplementary section.

Specific comments: Regarding the style of writing and choice of words:

  1. Line 51: Here and throughout the manuscript, please, use either the version “E4031” or “E-4031” not both.

Response: Granted. We changed all to a single denomination: E4031.

  1. Line 51: I would consider starting a new paragraph from the sentence “As acquired long QT is a life-threatening…”. In its current form the paragraph is exhaustively long.

Response: yes, good idea. Done.

  1. Line 54: Maybe write “previously” instead of “initially”?

Response: yes, done.

  1. Line 85: I would consider starting a new paragraph from the sentence “Herein, we now described the proarrhythmic…”.

Response: We agree.

  1. Lines 85-88: This is a looong sentence: “Herein, we now described the proarrhythmic effects of BeKm-1 on non-commercial hiPS-CMs using a variety of approaches and conclude that a block through the outer pore does not constitute enough a differentiating factor compared to other classical blocker in investigating the role of hERG channels.” Please, consider rewriting it for example like this: “Herein, we characterize the proarrhythmic effects of BeKm-1 on non-commercial hiPS-CMs using a variety of approaches. We conclude that a block through the outer pore does not constitute enough a differentiating factor compared to other classical blocker in investigating the role of hERG channels.”

Response: we modified this sentence. We thank the reviewer for this suggestion.

  1. Lines 105-106: Think “lower” should be “higher”, right? Or am I missing something here?

Response: No, the reviewer is right. We thank him/her for noticing it. We modified it.

  1. Line 162: Why “again”? To what is this “again” referring to?

Response: we removed it.

  1. Line 170: I would consider starting a new paragraph from the sentence “Although the effect of BeKm-1 on APD90 was expected…”.

Response: Done.

  1. Lines 178-179: The following sentence is written kind of backwards, in my opinion: “This observation is coherent with the fact that IK1 current is required to maintain the membrane potential in hiPS-CMs.” Maybe it could be rewritten like this: “This observation is coherent with the fact that a stronger IK1 current would be needed in hiPS-CMs to maintain a more negative membrane potential typical for CMs.”

Response: Perfect suggestion. Adopted.

  1. Line 181: I would write “can simulate” instead of “mimics”.

Response: Yes, fine.

  1. Line 184: “AP” instead of “Ap”.
  2. Lines 188-189: Second half of the sentence is a bit cumbersome. I would write: “…at -80 mV in these hiPS-CMs, causing the cessation of spontaneous APs (not shown).”

Response: OK for these two modifications.

  1. Lines 232-234: Weird sentence structure. I would write instead: “As saturating concentrations of BeKm-1 (100 nM) induced arrest of spontaneous APs in individual cells, we were…”

Response: Yes, fine.

  1. Line 247: In my opinion the subheading is misleading. That is, BeKm-1 does not alter Ca2+ handling Instead, it affects indirectly the intracellular Ca2+ dynamics. So, a proper subheading would be something like “BeKm-1 alters the Ca2+ dynamics in hiPS-CMs”.

Response: We agree. We added the word “indirectly” in the suggested subheading.

  1. Line 257: Weird choice of words: “decreases Ca2+ handling in the cytoplasm”. Write instead, for example, “Ca2+ transient amplitude in the cytoplasm”. On a related note, please, consider introducing the abbreviation “CaT” or “CT” for “Ca2+ transient” and use it throughout the manuscript.

Response: Actually, what is measured here is not the amplitude of the Ca transient. It is the integral of the response. We rephrased the sentence, but we cannot use the term amplitude. We now introduced this CaT abbreviation throughout where possible.

  1. Lines 263-264: Same comment here as on line 247. BeKm-1 does not alter Ca2+ handling. Instead, it affects indirectly the intracellular Ca2+ dynamics.

Response: yes true. Modified.

  1. Line 295: What is the meaning of the “*” sign in Figure 6E?

Response: this is statistical significance at p<0.05. Now added in figure legend.

  1. Line 326: I don’t understand, what “reduction of ECC” means. Please, consider rewriting that.

Response: Rewritten.

  1. Line 326: “Ca2+ transient” instead of “transient”.

Response: Yes indeed.

  1. Line 327: I would start a new paragraph from “We reported proarrhythmic effects…”. That way the Discussion section would not be one single, exhaustively long paragraph.

Response: This is fine with us.

  1. Line 348: I’m not sure, what the authors mean to say here: “…demonstrating that BeKm-1 may interestingly recapitulate some pathological properties of hiPS-CMs.” Maybe you could write instead: ““…demonstrating that BeKm-1 can induce pathological conditions in hiPS-CMs.”

Response: Yes, this is more explicit indeed.

  1. Lines 385-390: This is a quite vague description: “Between 2 and 2.8 pA/pF in silico IK1 density was generally sufficient to bring resting membrane potential (RMP) to -80 ± 5 mV. Cells whose RMP didn’t reach these physiological ventricular values were excluded from the recording process. During dynamic patch-clamp, cells were depolarized every 700 ms with a 30-35 pA/pF current density pulse. This current density was set in such a manner that not all ventricular cardiomyocytes were able to reach an RMP value of -80 mV.” Could you be a bit more specific/quantitative about your approach? Also, I guess “ventricular cardiomyocytes” should be “ventricular-like hiPS-CMs”, right?

Response: We modified this paragraph to be more descriptive. Yes “ventricular-like” as it is not an absolute assessment.

  1. Line 473: Weird choice of words. I would write “contrary to” instead of “at odd with”.

Response: Granted. We thank the reviewer for these extended corrections.

Reviewer 2 Report

This study is written well, however this need to be revised a few points.

1) Abstract; Authors described that his study was meant to assess the 24 modification in hiPS-CMs excitability and contractile properties by BeKm-1, a natural scorpion 25 venom peptide that selectively interacts with the extracellular face of hERG. However, I did not understand this hypothesis, so please describe this. 

2) Introduction; same as this abstract.

Author Response

Reviewer 2

This study is written well, however this need to be revised a few points.

1) Abstract; Authors described that his study was meant to assess the 24 modification in hiPS-CMs excitability and contractile properties by BeKm-1, a natural scorpion 25 venom peptide that selectively interacts with the extracellular face of hERG. However, I did not understand this hypothesis, so please describe this. 

2) Introduction; same as this abstract.

Response: We thank the reviewer for this question. In assessing the effect of a drug on human cardiomyocytes, all reference compounds generally act from the inside of the cell and hit the intracellular face of the channel. Yet, unknown compounds should not necessarily act in similar ways then these reference compounds, but could as well act by binding onto the extracellular face of the channel. Therefore, BeKm-1 might become an interesting or maybe better reference compound for an adequate comparison. We tried to be more explicit in the introduction on this issue.